# Development of at-home sample collection logistics for large-scale seroprevalence studies

**Aishani V. Aatresh**[1,2], **Kate Cummings**[3], **Hilary Gerstein**[3], **Christopher S. Knight**[3], **Andreas Limberopolous**[3], **Megan A. Stasi**[3], **Alice Bedugnis**[1], **Kenneth A. Somberg**[3], **Camila T. França**[1☯]*, **Michael J. Mina**[1,4,5☯]*

**1** Department of Epidemiology, Center for Communicable Disease Dynamics, Harvard T.H. Chan School of Public Health, Boston, MA, United States of America, **2** Harvard College, Cambridge, MA, United States of America, **3** TrialSpark, Inc., New York, NY, United States of America, **4** Department of Immunology and Infectious Diseases, Harvard T.H. Chan School of Public Health, Boston, MA, United States of America, **5** Department of Pathology, Brigham and Women's Hospital and Harvard Medical School, Boston, MA, United States of America

☯ These authors contributed equally to this work.
* cfranca@hsph.harvard.edu (CTF); mmina@hsph.harvard.edu (MJM)

**Data Availability Statement:** Data has not been made publicly available to avoid compromising participant privacy or violating the ethical agreement in the informed consent form. Data

## Abstract

### Background

Serological studies rely on the recruitment of representative cohorts; however, such efforts are specially complicated by the conditions surrounding the COVID19 pandemic.

### Methods

We aimed to design and implement a fully remote methodology for conducting safe serological surveys that also allow for the engagement of representative study populations.

### Results

This design was well-received and effective. 2,066 participants $\geq$18 years old were enrolled, reflecting the ethnic and racial composition of Massachusetts. >70% of them reported being satisfied/extremely satisfied with the online enrollment and at-home self-collection of blood samples. While 18.6% reported some discomfort experienced with the collection process, 72.2% stated that they would be willing to test weekly if enrolled in a long-term study.

### Conclusions

High engagement and positive feedback from participants, as well as the quality of self-collected specimens, point to the usefulness of this fully remote, self-collection-based study design for future safer and efficient population-level serological surveys.

might be made available upon reasonable request by contacting Prof. Michael J. Mina or Alyssa Pellegrini (apellegrini@hsph.harvard.edu).

**Funding:** This study was funded by Open Research. M.J.M. is supported by the NIH Director's Early Independence Award DP5OD028145. he funders had no role in study design, data collection and analysis, decision to publish, or preparation of the manuscript.

**Competing interests:** M.J.M. is supported by the NIH Director's Early Independence Award DP5OD028145. The authors K.C., H.G., C.S.K, A.L, M.A.S, and K.A.S. are employees of TrialSpark, Inc. This study was funded by Open Research. The funders had no role in study design, data collection and analysis, decision to publish, or preparation of the manuscript. The conflicts declared above do not alter our adherence to PLOS ONE policies on sharing data and materials.

## Introduction

The coronavirus disease 2019 (COVID-19) pandemic caused by severe acute respiratory syndrome coronavirus 2 (SARS-CoV-2) has had far-reaching consequences since its emergence in Wuhan, China, in December 2019 [1]. As of July 2021, even though vaccination has become increasingly widespread, there have been over 191 million cases and 4.11 million deaths accounted for worldwide. The more subtle cost exacted upon society has been evident in the rise of virtual school, remote work, severe job loss, and economic contraction [2].

Studies surrounding the humoral response mounted against SARS-CoV-2 infection continue to emerge as the pandemic persists [3–5]. As the measurement of antibodies against SARS-CoV-2 in blood is relatively cheap, serology has been proposed as an alternative method to identify individuals who have previously had symptomatic or asymptomatic SARS-CoV-2 infections and recovered [6]. Useful not only for COVID-19-related studies, well-designed population sero-surveys can be powerful tools to help determine trend of diseases [7]. Such studies can also provide a better understanding of the dynamics of antibody responses for differentiation of individuals with acquired immunity from those who remain susceptible to infection and disease, therefore helping to determine where to deploy resources for disease prevention and management, and helping identify emerging outbreaks early [8].

In order to facilitate the use of serology as a public health tool, we aimed to design and implement a fully remote mechanism for conducting large-scale serosurveys. We coupled the use of electronic medium for study engagement and successful recruitment and retention of representative cohorts with at-home self-collection of serological specimens using fingerpick collection, allowing for increased sampling of diverse populations with better efficiency and cost and significantly greater participant safety.

We implemented these logistics by successfully conducting a large cross-sectional survey of the population of Massachusetts and measuring the prevalence of total IgG antibodies to SARS-CoV-2 in symptomatic and asymptomatic individuals. Findings provide a proof-of-concept for the logistics for safer sero-epidemiological studies.

## Methods

### Study design

**Ethics approval.** Ethical clearance was obtained from Advarra (Pro00043729) and the Harvard T.H. Chan School of Public Health review board (IRB20-1511). Written informed consent was obtained electronically from all participants prior to enrollment in this study.

**Recruitment.** This at-home, decentralized study targeted adult (≥18 years of age) residents of Massachusetts. With the goal of enrolling approximately 2,000 volunteers, potential participants were identified through partnerships with for- and non-profit entities and digital ad campaigns and referrals and received a link to a landing page to learn more about the study and enroll if interested. Participants were required to have reliable Internet access and to speak English, as the study was not offered in additional languages. If eligible, participants electronically reviewed the informed consent form and completed a background questionnaire (S1 Table) about their demographic profile (including gender, age, race, ethnicity, residency, education, income, housing status, pregnancy and recent medical history/comorbid conditions), and COVID-19 history (including presumptive and confirmed SARS-CoV-2, checklist of symptoms and their duration, level of care received and clinical outcome, adherence to social distancing guidelines, use of masks/face coverings in public and type of transportation used). Volunteers were not compensated for their participation in the study.

**Specimen collection.** After completing the baseline questionnaire, participants were shipped through the United States Postal Service (USPS) an at-home specimen collection kit which included two spring-loaded lancets, a biohazard bag, and instructions for self-administered finger-prick blood collection. Participants were asked to place approximately 10–20 drops of blood onto the supplied Whatman 903 dried blood spot protein saver filter paper. After air drying the specimen, the participants were instructed to place the filter paper into sealed, pre-paid envelopes provided in the kit and mail it to Molecular Testing Labs, a Clinical Laboratory Improvement Amendments (CLIA)-licensed laboratory, for analysis. All participants with a positive SARS-CoV-2 IgG result were asked to provide additional blood finger-prick samples at day 7, 15, 45, and 90 after receiving the initial result. Throughout the study, all participants had access to frequently asked questions, as well as a dedicated support team and nurse online or by phone.

**Laboratory tests.** The presence of total IgG antibodies against the S1 protein of SARS-CoV-2 was measured using the EUROIMMUN ELISA assay as previously described [9]. Test results were returned to the participants by Molecular Testing Labs within 24–72 hours of receipt of the specimen using the study mobile application platform as positive, negative, or indeterminate. A second kit was offered to any participant who received an indeterminate result and wished to provide another specimen.

**Statistical analysis.** Chi-square tests were used to investigate the association between demographic, clinical, and behavioral factors and seroprevalence of antibodies against SARS-CoV-2. All analyses were performed using Python (version 3.8.5).

## Results

### Study enrollment and participant demographics

690 of the planned 2,000 participants were enrolled in only two weeks (June 16–30, 2020) using convenience sampling (Fig 1A and 1B). Most of this initial population was comprised of Caucasian, high-income (>$140,000) individuals (Table 1). In order to increase diversity to mirror race and ethnicity proportions of 2019 Massachusetts census data and achieve a 50/50 split between residency within rural or urban centers (as defined by the Massachusetts State Office of Rural Health based on population size and density, hospital availability, and the Census Bureau and Office of Management and Budget), age, zip code, internet access, race and ethnicity information were used to pre-screen interested individuals and temporarily place them on a waiting list/lottery. The remaining participants (n = 1,376) were enrolled between July 29—August 24, 2020. In total, 48.3% (n = 939) of participants for whom recruitment data was available (n = 1,945) were recruited through online ads, specifically Facebook (Fig 1B).

From an initial cohort of 2,066 study participants, 90.61% (n = 1872) of individuals mailed their sample to the laboratory for analysis. The median age of study enrollees was 40 years old (interquartile range [IQR] 32 to 52 years old), 73.95% (n = 1368) were female, while 81.37% (n = 1681) hold an undergraduate degree or higher (Table 1). The cohort was generally distributed over the state of Massachusetts, with 48.65% and 51.35% from rural and urban areas, respectively. A total of 40.11% (n = 742) reported having symptoms resembling those of COVID-19 since January 2020 (including cough, fever, shortness of breath, sore throat, and new loss of smell or taste) and 14.09% (n = 291) reported having one or more comorbid health conditions known to increase risk of COVID-19 (e.g., diabetes, asthma, being immunocompromised, heart or lung disease) (Table 1).

### SARS-CoV-2 serology in Massachusetts

Using our at-home self-collection logistics, 3.15% (n = 59) of the individuals who returned their samples were seropositive for total IgG antibodies against SARS-CoV-2 S1 protein

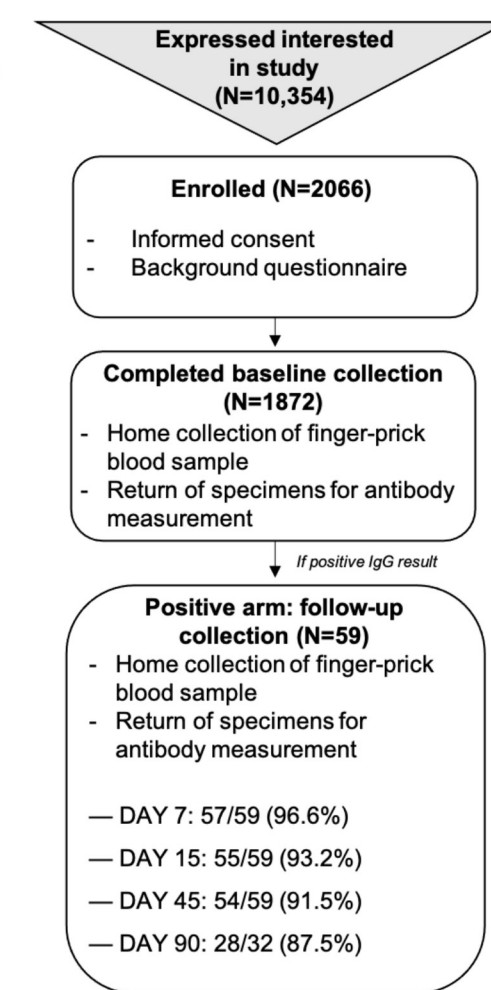

a)

**Expressed interested in study (N=10,354)**

**Enrolled (N=2066)**
- Informed consent
- Background questionnaire

**Completed baseline collection (N=1872)**
- Home collection of finger-prick blood sample
- Return of specimens for antibody measurement

*If positive IgG result*

**Positive arm: follow-up collection (N=59)**
- Home collection of finger-prick blood sample
- Return of specimens for antibody measurement

— DAY 7: 57/59 (96.6%)

— DAY 15: 55/59 (93.2%)

— DAY 45: 54/59 (91.5%)

— DAY 90: 28/32 (87.5%)

b)

| Source | Enrolled Participants (Count) |
|---|---|
| Facebook ads | 939 |
| PatientPing | 379 |
| Outreach | 182 |
| Direct contact/ No source | 260 |
| COVID Symptom Study | 152 |
| Quinsigamond Community College | 29 |
| Flu Near You | 4 |

**Fig 1. Study design.** a) Progression of study from recruitment and participant admission to testing and follow-up sample collection for individuals who test positive for IgG antibodies against SARS-CoV-2 at baseline. b) Counts of study population garnered through each method; known sources and counts of participants recruited electronically across the state of Massachusetts.

(S1A Fig) and were requested to send follow-up samples at days 7, 14, 45, and 90 after initial positive result (S1B Fig). Out of the 59 participants with baseline positive results, the vast majority remained seropositive, with 7.27% (n = 4 of 55) showing indeterminate, and 16.36% (n = 9 of 55) seronegative results by Day 90 (S1B Fig). A higher risk of infection was observed for symptomatic individuals (p<0.001), as well as those of lower-income (p = 0.03), less educated (p<0.001), Hispanic (p = 0.02), and those in the age groups of 18–29 and 50-59-years-old (p = 0.02); sample size limitation of different groupings prevent the same conclusion from being made about the risk of infection based on the number of individuals in a household (p<0.001) (S2 Table, S1C Fig).

## Participant feedback

After sample collection and testing were finalized, 1,764 participants were sent a survey to provide feedback about the study process, eliciting a 31% (n = 547) overall response rate. Survey respondents were generally representative of the study population (S2 Fig). 96.16% (n = 526) of them reported being extremely satisfied or satisfied with the process of enrolling in the

**Table 1. Demographic profile of study participants.**

| Category | Cohort | Cohort Goals | | Actual Enrolled | |
|---|---|---|---|---|---|
| | | # | % (n = 2000) | # | % (n = 2066) |
| *Race (not mutually exclusive, includes double counting of 2+ races which precludes % from summing to 100%)* | Black or African American | 180 | 9.00% | 188 | 9.10% |
| | Asian | 144 | 7.20% | 171 | 8.30% |
| | Other non-white (includes 2 or more races) | 64 | 3.20% | 207 | 10.00% |
| | White | 1612 | 80.60% | 1696 | 82.10% |
| *Ethnicity* | Hispanic or Latinx | 248 | 12.40% | 275 | 13.30% |
| | Not Hispanic or Latinx | 1752 | 87.60% | 1792 | 86.70% |
| *Rural vs. Urban* | Rural | 1000 | 50.00% | 1000 | 48.40% |
| | Urban | 1000 | 50.00% | 1067 | 51.60% |
| *Age* | 18–29 | | N/A | 362 | 17.55% |
| | 30–39 | | N/A | 621 | 10.95% |
| | 40–49 | | N/A | 484 | 8.54% |
| | 50–59 | | N/A | 344 | 6.07% |
| | 60–69 | | N/A | 206 | 3.63% |
| | 70–79 | | N/A | 41 | 0.72% |
| | 80–89 | | N/A | 5 | 0.09% |
| *Gender* | Male | | N/A | 504 | 24.39% |
| | Female | | N/A | 1536 | 74.35% |
| | Other | | N/A | 26 | 12.58% |
| *Comorbidities* | None | | N/A | 1775 | 87.01% |
| | One or more | | N/A | 291 | 14.26% |
| *Education* | Graduate or professional | | N/A | 928 | 44.92% |
| | Bachelor's | | N/A | 753 | 36.44% |
| | Some college or associate's | | N/A | 321 | 15.54% |
| | High school graduate/GED | | N/A | 59 | 2.86% |
| | Some high school/did not attend | | N/A | 5 | 0.24% |
| *Income* | $140,000 + | | N/A | 738 | 35.50% |
| | $45,000 — $139,999 | | N/A | 305 | 14.67% |
| | $100,000 — $139,999 | | N/A | 264 | 12.70% |
| | $75,000 — $99,999 | | N/A | 209 | 10.05% |
| | $50,000 — $74,999 | | N/A | 178 | 8.56% |
| | $20,000 — $49,999 | | N/A | 173 | 8.32% |
| | Less than $20,000 | | N/A | 58 | 2.79% |
| | Prefer not to answer | | N/A | 141 | 6.78% |

study (Fig 2A), 84.68% (n = 459) reported being extremely satisfied or satisfied with the experience of self-collection of the finger-prick sample (Fig 2B), with the majority of responses indicating sample collection was very to extremely easy (Fig 2D). More respondents rated the experience as more comfortable than not (Fig 2E). Meanwhile, 95.37% (n = 515) were extremely satisfied or satisfied with the content and quality of study communications (Fig 2C). With respect to the potential for future studies, 56.67% (n = 306) of respondents said they were extremely likely to recommend this method of remote enrollment, at-home self-collection of specimens and antibody testing to others (Fig 2F). 72.23% (n = 385) of the responders were willing to self-perform finger-prick blood collection up to once per week if needed (Fig 2G and 2H). While 63.65% (n = 345) of the respondents did not have children, 19.37% (n = 105) of those who did indicate that they would enroll their child in such a study (Fig 2I).

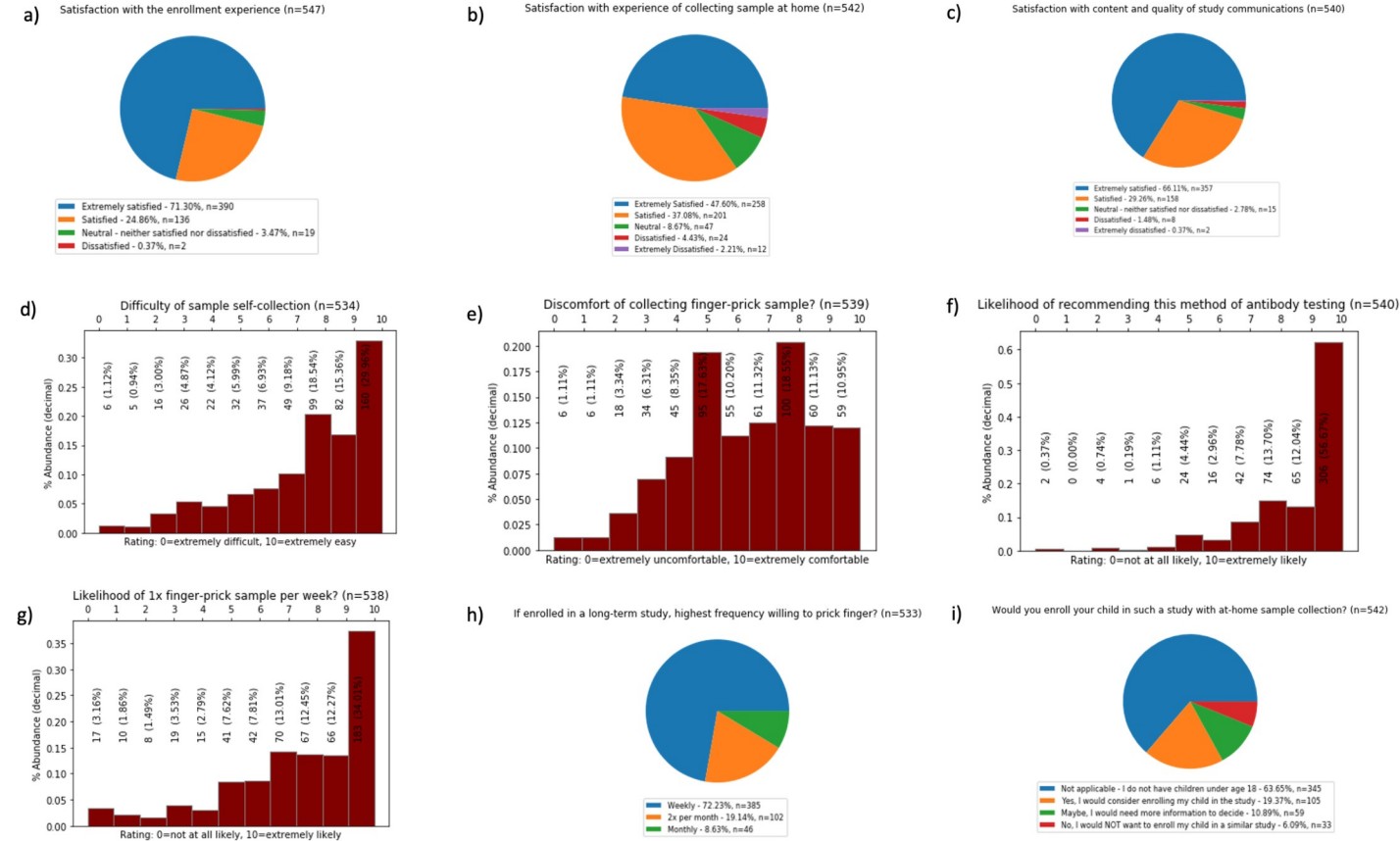

**Fig 2. Participant feedback.** Distribution of the participants who responded the feedback survey by satisfaction with study participation (a, b, c), difficulty with self-collection of blood sample (d, e), and willingness to collect samples at higher frequency, recommend this study to others or enroll their child(ren) in similar studies (f, g, h, i).

## Discussion

Given the COVID-19 pandemic, the risk of viral transmission and limited capacity of health-care systems called for the decentralized, at-home nature of this seroprevalence study, leveraging online recruitment, eConsent, electronic questionnaires, and direct-to-patient shipping to reach a broad representative study population. This study was a valuable opportunity to utilize and assess an at-home approach, and participant survey data reveals it was overwhelmingly well-received and indicates a strong likelihood of success for future deployment of larger studies of this modality. Although the discomfort of the finger-prick was the biggest concern expressed by participants, self-collection of samples was reported to be easy and generated samples of quality without the need for trained professionals or personal protective equipment, providing a remedy for the difficulties often encountered when obtaining standard specimens by phlebotomy, particularly during a pandemic.

While representative cohorts are especially important for COVID-19 prevalence estimation because of the disproportionate impact of that this pandemic has exacted upon racial and ethnic minorities [10], minimally biased data regarding the status of the pandemic has been significantly limited thus far [11–14]. Convenience sampling can skew data by drawing a study cohort that is not representative of the underlying population, as such surveys may not be able to adequately reach less advantaged communities, whether in rural areas or in lower-income

urban settings. Furthermore, individuals seeking or willing to receive testing may be more likely to have experienced illness.

The recruitment strategy employed in the present study was very successful in reaching a representation of the population structure seen across Massachusetts with respect to race, ethnicity, and location of residency. However, recruitment was still subject to skew towards individuals who were more prolific Facebook users, female, highly educated, and wealthy (>$140,000 annual income). A small number of participants were not fully random because of shared households and thus could be non-independent exposures.

Therefore, while the general utility and receptiveness of the method applied here is demonstrated, future studies would benefit from restricting general engagement and recruitment to focus on populations living in specific disease-burdened areas or from specific income and education levels.

At the time of this study in July-August of 2020, the incidence of COVID-19 antibodies in the population enrolled in this study was relatively low (3.15%). Individuals in the age ranges of 18 to 29 and 50 to 59 years were more likely to have antibodies to SARS-CoV-2, which likely reflect behavioral patterns (e.g., possibly less careful social behavior) and increased transmission among the young and a covarying increased risk of disease by greater incidence of comorbidities among the older, respectively. For the 59 of 1872 individuals with positive IgG to SARS-CoV-2 at baseline, sustained serological responses are generally observed, with the subset of negative results at day 90 possibly serving as an indicator of the natural waning of an antibody response over time [15]. It is important to highlight that some of the samples may appear as a false negative or a false positive also because of the limitations of the commercial test used (EUROIMMUN), as sensitivity and specificity are predicted to be lower in low prevalence settings [9].

Other studies aiming to conduct similarly remote serology did not assess participant satisfaction and rather focused on seroprevalence or technological performance [16–18]. The focus of our study was however, to interrogate the nature of participants' willingness to engage in such study design and their overall experience in order to provide a more comprehensive understanding of the usefulness and role remote serology studies stand to play in infectious disease surveillance. The seroprevalence results we found were in accordance to the overall pattern developing in the USA during July-August 2020 and reported by studies conducted using traditional (assisted) sample collection [19, 20].

Participants were certainly influenced by the climate surrounding the pandemic and the public health measures implemented, such as stay-at-home orders and social distancing, and thus were likely to have been more inclined to partake in a remote surveillance study. While the incentive to do so in a non-pandemic world could be reduced, the low amount of sample needed and reduced pain in comparison to traditional venipuncture approaches that require in-person phlebotomy visits are benefits that might retain interest in participating in future remote studies. Importantly, such a remote design would facilitate collection of samples in very large scale (eg nationwide) without the need for very large teams for community engagement and sample collection, therefore being potentially useful in effectively integrating public health pathogen surveillance (i.e. "peacetime" surveillance) into day-to-day practices of community and individual health, hopefully viewed as a worthy cause in which individuals participate to avert future pandemics [21].

## Supporting information

**S1 Table. Participant profile questionnaire.** Background questionnaire completed by participants after informed consent was obtained, gathering self-reported demographic and clinical

data.
(XLSX)

**S2 Table. Demographic breakdown of SARS-CoV-2 seropositivity in Massachusetts.**
(XLSX)

**S1 Fig. Seroprevalence of SARS-CoV-2 total IgG antibodies in Massachusetts.** a) Distribution of positive, negative and indeterminate results for presence of total IgG antibodies against SARS-CoV-2 S1 protein across all individuals who returned a baseline test specimen (n = 1872). b) Heatmap showing presence of total IgG antibodies against SARS-CoV-2 S1 protein in follow-up samples of individuals who tested positive at baseline. Each row represents an individual and each column a time-point of sample collection (baseline, days 7, 15, 45 and 90) with data complete as of March 3, 2021. c) Histograms showing the total counts (left y-axis) for each variable in the study population. Black crosses represent percentage seropositivity (right y-axis) against the entire population (n = 1872) given individuals for each group.
(TIF)

**S2 Fig. Representative survey sample.** Comparison of general distribution of survey sample (maroon, n = 542) against general sample distribution of study population (green, n = 2066/as data is available for age, n = 2063) for the commonly collected demographic variables of income, ethnicity, and age.
(TIF)

## Acknowledgments

We are grateful to all the volunteers who participated in this study. We would like to thank the TrialSpark Clinical Research Coordinators and Recruitment, Operations, Data, Marketing and Technology teams for partnering on the development and successful deployment of the logistics used in this study. We thank the Molecular Testing Lab for IgG measurements.

## Author Contributions

**Conceptualization:** Kenneth A. Somberg, Camila T. França, Michael J. Mina.

**Data curation:** Alice Bedugnis.

**Formal analysis:** Aishani V. Aatresh.

**Funding acquisition:** Kenneth A. Somberg, Michael J. Mina.

**Investigation:** Aishani V. Aatresh, Alice Bedugnis, Camila T. França.

**Methodology:** Kate Cummings, Hilary Gerstein, Christopher S. Knight, Camila T. França, Michael J. Mina.

**Project administration:** Camila T. França.

**Resources:** Kate Cummings, Hilary Gerstein, Christopher S. Knight.

**Software:** Andreas Limberopolous.

**Supervision:** Megan A. Stasi, Camila T. França, Michael J. Mina.

**Visualization:** Aishani V. Aatresh.

**Writing – original draft:** Aishani V. Aatresh, Camila T. França.

**Writing – review & editing:** Aishani V. Aatresh, Kenneth A. Somberg, Camila T. França, Michael J. Mina.

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
