## [Decision Letter · Decision Letter 0]

3 Sep 2021

PONE-D-21-24248

Development of at-home sample collection logistics for large-scale seroprevalence studies

PLOS ONE

Dear Dr. Aatresh,

Thank you for submitting your manuscript to PLOS ONE. After careful consideration, we feel that it has merit but does not fully meet PLOS ONE’s publication criteria as it currently stands. Therefore, we invite you to submit a revised version of the manuscript that addresses the points raised during the review process.

We look forward to receiving your revised manuscript.

Kind regards,

James Drummond

Academic Editor

PLOS ONE

Journal Requirements:

2. Thank you for declaring your affiliation with Trial Spark, Inc. In the competing interests statement within the manuscript and in the online submission form, please clarify the nature of this competing interest.

3. Thank you for stating the following in the Competing Interests/Financial Disclosure * (delete as necessary) section: 

"Conflict of Interest

K.C., H.G., C.S.K, A.L, M.A.S, and K.A.S. are employees of TrialSpark, Inc." 

We note that you received funding from a commercial source: TrialSpark, Inc.

5. We note that Supplementary Figure 1 in your submission contain [map/satellite] images which may be copyrighted. All PLOS content is published under the Creative Commons Attribution License (CC BY 4.0), which means that the manuscript, images, and Supporting Information files will be freely available online, and any third party is permitted to access, download, copy, distribute, and use these materials in any way, even commercially, with proper attribution. For these reasons, we cannot publish previously copyrighted maps or satellite images created using proprietary data, such as Google software (Google Maps, Street View, and Earth). For more information, see our copyright guidelines: http://journals.plos.org/plosone/s/licenses-and-copyright.

a. You may seek permission from the original copyright holder of Supplementary Figure 1 to publish the content specifically under the CC BY 4.0 license.  

Additional Editor Comments:

In order to make this article more useful to others in the field

Please complete all of the edits suggested by the reviewers. See the reviewers comments for more detail.

Briefly:

Ensure that the percentages in Table 1 are calculated correctly

Include the questionnaire, at least linked to in supplemental material

Present the results of the ELISA tests in a table/graph.

Discuss whether results are similar, or not, with other reported results from different collection/testing methodologies?

Reviewers' comments:

Reviewer's Responses to Questions

**Comments to the Author**

1. Is the manuscript technically sound, and do the data support the conclusions?

Reviewer #1: Yes

Reviewer #2: Yes

2. Has the statistical analysis been performed appropriately and rigorously? 

Reviewer #1: Yes

Reviewer #2: No

3. Have the authors made all data underlying the findings in their manuscript fully available?

Reviewer #1: Yes

Reviewer #2: No

4. Is the manuscript presented in an intelligible fashion and written in standard English?

Reviewer #1: Yes

Reviewer #2: Yes

5. Review Comments to the Author

Reviewer #1: This is a relatively simple but interesting study, the fully remote methodology for conducting safe serological surveys is especially meaningful during the current pandemic, and the logistics seem readily applicable for other sero-epidemiological studies.

The only concern that I have is that the serology results (actual Figure 2) is completely omitted in the Results and Discussion sessions, while the Figure 2 in current Results session should actually be Figure 3 (participant feedback). Although it's understandable that the serology results not being the focal point of this study, they still need to be reported at least in the Results session to demonstrate the validity of the methodology.

Reviewer #2: This descriptive study by Aatresh et al. outlines the logistics of at-home sample collection for seroprevalence studies using a combination of online data collection tools and a fingerpick collection kit. This study exploited the need for at-home testing services for COVID-19 to examine the feasibility of this system of data collection for other types of studies. This is an interesting proof-of-concept, and provides useful data for future studies that would propose this type of data collection for the convenience of ascertaining large numbers of samples or frequency data points on a cohort. Several points of clarification are needed as outlined below.

1. The percentages computed in Table 1 are not correct (some of the cells have the wrong denominator apparently). Some of the stratified percentages do not sum to 100.

2. Some of the demographic variables are not well-defined in the methods. What criteria are used to define "urban" versus "rural"? Were the comorbidities an open response or a selected set of responses?

3. It would be helpful to see the exact phrasing of each question in the questionnaire as part of the supplemental materials. This would leave no ambiguity about the potential interpretations of the responses.

4. The major utility of this work is its potential to serve as a general view of the logistic set-up used for this study. A significant concern that should be addressed in the discussion is: How likely are these results, collected as part of an infectious disease surveillance study in the middle of a pandemic, likely to generalize to studies post-pandemic? In other words, are the participation rates for the demographic categories mentioned, skewed by the overwhelming desire to stay safe during the pandemic? It is quite possible that participation rates or viewpoints may change outside the current environment. This should at least be addressed in the discussion.

6. PLOS authors have the option to publish the peer review history of their article (what does this mean?). If published, this will include your full peer review and any attached files.

Reviewer #1: No

Reviewer #2: No

---

## [Author Response · Author response to Decision Letter 0]

23 Sep 2021

Development of at-home sample collection logistics for large-scale seroprevalence studies

Responses to the editor’s comments:

Thank you. We have revised and corrected our files to meet PLOS ONE’s requirements.

2. Thank you for declaring your affiliation with Trial Spark, Inc. In the competing interests statement within the manuscript and in the online submission form, please clarify the nature of this competing interest.

We have included the following in the competing interests section of the manuscript:

“M.J.M. is supported by the NIH Director’s Early Independence Award DP5OD028145. The authors K.C., H.G., C.S.K, A.L, M.A.S, and K.A.S. are employees of TrialSpark, Inc. This study was funded by Open Research. The funders had no role in study design, data collection and analysis, decision to publish, or preparation of the manuscript. The conflicts declared above do not alter our adherence to PLOS ONE policies on sharing data and materials.”

3. Thank you for stating the following in the Competing Interests/Financial Disclosure * (delete as necessary) section: "Conflict of Interest K.C., H.G., C.S.K, A.L, M.A.S, and K.A.S. are employees of TrialSpark, Inc." We note that you received funding from a commercial source: TrialSpark, Inc. Please provide an amended Competing Interests Statement that explicitly states this commercial funder, along with any other relevant declarations relating to employment, consultancy, patents, products in development, marketed products, etc. Within this Competing Interests Statement, please confirm that this does not alter your adherence to all PLOS ONE policies on sharing data and materials by including the following statement: "This does not alter our adherence to PLOS ONE policies on sharing data and materials.” (as detailed online in our guide for authors http://journals.plos.org/plosone/s/competing-interests). If there are restrictions on sharing of data and/or materials, please state these. Please note that we cannot proceed with consideration of your article until this information has been declared. Please include your amended Competing Interests Statement within your cover letter. We will change the online submission form on your behalf.

Thank you. We have amended our Conflict of interest/financial disclosure section, as well as our cover letter with the following:

“M.J.M. is supported by the NIH Director’s Early Independence Award DP5OD028145. The authors K.C., H.G., C.S.K, A.L, M.A.S, and K.A.S. are employees of TrialSpark, Inc. This study was funded by Open Research. The funders had no role in study design, data collection and analysis, decision to publish, or preparation of the manuscript. The conflicts declared above do not alter our adherence to PLOS ONE policies on sharing data and materials.”

Our ethics approval statement (below) has been moved to the Methods section:

“Ethical clearance was obtained from Advarra (Pro00043729) and the Harvard T.H. Chan School of Public Health review board (IRB20-1511). Written informed consent was obtained electronically from all participants prior to enrollment in this study.”

5. We note that Supplementary Figure 1 in your submission contain [map/satellite] images which may be copyrighted. All PLOS content is published under the Creative Commons Attribution License (CC BY 4.0), which means that the manuscript, images, and Supporting Information files will be freely available online, and any third party is permitted to access, download, copy, distribute, and use these materials in any way, even commercially, with proper attribution. For these reasons, we cannot publish previously copyrighted maps or satellite images created using proprietary data, such as Google software (Google Maps, Street View, and Earth). For more information, see our copyright guidelines:http://journals.plos.org/plosone/s/licenses-and-copyright. We require you to either (1) present written permission from the copyright holder to publish these figures specifically under the CC BY 4.0 license, or (2) remove the figures from your submission: a. You may seek permission from the original copyright holder of Supplementary Figure 1 to publish the content specifically under the CC BY 4.0 license. We recommend that you contact the original copyright holder with the Content Permission Form (http://journals.plos.org/plosone/s/file?id=7c09/content-permission-form.pdf) and the following text: “I request permission for the open-access journal PLOS ONE to publish XXX under the Creative Commons Attribution License (CCAL) CC BY 4.0 (http://creativecommons.org/licenses/by/4.0/). Please be aware that this license allows unrestricted use and distribution, even commercially, by third parties. Please reply and provide explicit written permission to publish XXX under a CC BY license and complete the attached form. Please upload the completed Content Permission Form or other proof of granted permissions as an "Other" file with your submission. In the figure caption of the copyrighted figure, please include the following text: “Reprinted from [ref] under a CC BY license, with permission from [name of publisher], original copyright [original copyright year]. b. If you are unable to obtain permission from the original copyright holder to publish these figures under the CC BY 4.0 license or if the copyright holder’s requirements are incompatible with the CC BY 4.0 license, please either i) remove the figure or ii) supply a replacement figure that complies with the CC BY 4.0 license. Please check copyright information on all replacement figures and update the figure caption with source information. If applicable, please specify in the figure caption text when a figure is similar but not identical to the original image and is therefore for illustrative purposes only. The following resources for replacing copyrighted map figures may be helpful:

We thank the editor for pointing this out and for all the suggestions. We were unable to obtain permission from the original copyright holder and therefore have decided to remove the figure in question (Supplementary figure 1) from our manuscript.

6. Ensure that the percentages in Table 1 are calculated correctly

This is a result of a software calculation error and has been corrected.

7. Include the questionnaire, at least linked to in supplemental material

We have included a copy of the questionnaire sent to the participants as Supplementary table 1

8. Present the results of the ELISA tests in a table/graph.

We have included our serological findings in the Results and Discussion sections (see below), as well as a supplementary figure (Supplementary figure 1):

Results: “Using our at home self-collection logistics, 3.15% (n=59) of the individuals who returned their samples were seropositive for total IgG antibodies against SARS-CoV-2 S1 protein at the time the study was conducted (Supplementary figure 1a) and were requested to send follow-up samples at days 7, 14, 45, and 90 after initial positive result (Supplementary figure 1b). Out of the 59 participants with baseline positive results, the vast majority remained seropositive, with 7.27% (n=4 of 55) showing indeterminate, and 16.36% (n=9 of 55) seronegative results by Day 90 (Supplementary figure 1b). A higher risk of infection was observed for symptomatic individuals (p<0.001), as well as those of lower-income (p=0.03), less educated (p<0.001), Hispanic (p=0.02), and those in the age groups of 18-29 and 50-59-years-old (p=0.02) (Supplementary figure 1c).”

Discussion: At the time of this study in July-August of 2020, the incidence of COVID-19 antibodies in the population enrolled in this study was relatively low (3.15%). Individuals in the age ranges of 18 to 29 and 50 to 59 years were more likely to have antibodies to SARS-CoV-2, which likely reflect behavioral patterns (e.g., possibly less careful social behavior) and increased transmission among the young and a covarying increased risk of disease by greater incidence of comorbidities among the older, respectively. For the 59 of 1872 individuals with positive IgG to SARS-CoV-2 at baseline, sustained serological responses are generally observed, with the subset of negative results at day 90 possibly serving as an indicator of the natural waning of an antibody response over time23. It is important to highlight that some of the samples may appear as a false negative or a false positive also because of the limitations of the commercial test used (EUROIMMUN), as sensitivity and specificity are predicted to be lower in low prevalence settings.

9. Discuss whether results are similar, or not, with other reported results from different collection/testing methodologies?

We have included a paragraph in the Discussion section addressing this comment: 

“Other studies aiming to conduct similarly remote serology studies did not assess participant satisfaction and rather focused on seroprevalence or technological performance [15], [16], [17]. The focus of our study was however, to interrogate the nature of participants’ willingness to engage with such study design and their overall experience in order to provide a more comprehensive understanding of the usefulness and role remote serology studies stand to play in infectious disease surveillance. The seroprevalence results we found were in accordance to the overall pattern developing in the USA during July-August 2020 reported by studies conducted using traditional (assisted) sample collection.”

Responses to reviewer #1 comments:

1. This is a relatively simple but interesting study, the fully remote methodology for conducting safe serological surveys is especially meaningful during the current pandemic, and the logistics seem readily applicable for other sero-epidemiological studies. The only concern that I have is that the serology results (actual Figure 2) is completely omitted in the Results and Discussion sessions, while the Figure 2 in current Results session should actually be Figure 3 (participant feedback). Although it's understandable that the serology results not being the focal point of this study, they still need to be reported at least in the Results session to demonstrate the validity of the methodology.

We thank reviewer 1 for his suggestion. This point is well taken. Although the specific serological results are not the focal point of this study, we included a brief section with these findings in Results (see below) and additional figure and table in the Supplementary section (supplementary figure 1) representing these results in more detail:

Results: “Using our at home self-collection logistics, 3.15% (n=59) of the individuals who returned their samples were seropositive for total IgG antibodies against SARS-CoV-2 S1 protein at the time the study was conducted (Supplementary figure 1a) and were requested to send follow-up samples at days 7, 14, 45, and 90 after initial positive result (Supplementary figure 1b). Out of the 59 participants with baseline positive results, the vast majority remained seropositive, with 7.27% (n=4 of 55) showing indeterminate, and 16.36% (n=9 of 55) seronegative results by Day 90 (Supplementary figure 1b). A higher risk of infection was observed for symptomatic individuals (p<0.001), as well as those of lower-income (p=0.03), less educated (p<0.001), Hispanic (p=0.02), and those in the age groups of 18-29 and 50-59-years-old (p=0.02) (Supplementary figure 1c, Supplementary Table 2).”

Responses to reviewer #2 comments:

1. The percentages computed in Table 1 are not correct (some of the cells have the wrong denominator apparently). Some of the stratified percentages do not sum to 100.

We apologize for this. This was a result of a software calculation error and has been corrected.

2. Some of the demographic variables are not well-defined in the methods. What criteria are used to define "urban" versus "rural"? Were the comorbidities an open response or a selected set of responses?

We used the definition “rural” vs “urban” as characterized by the Massachusetts State Office of Rural Health (https://www.mass.gov/service-details/state-office-of-rural-health-rural-definition). Factors such as population size and density, hospital availability, and the Census Bureau and Office of Management and Budget were used by the Massachusetts State Office of Rural Health to make these definitions. We have clarified this information in the Methods section.

Regarding the “Co-morbidities” variable, the participants were able to choose an answer from a selected set of responses. All of the co-morbidities known (at the time the study was conducted) to influence the risk of SARS-CoV-2 infection or severity were included. We have included a list with all questions and the response choices as Supplementary table 1.

3. It would be helpful to see the exact phrasing of each question in the questionnaire as part of the supplemental materials. This would leave no ambiguity about the potential interpretations of the responses.

We have included a list with all questions presented in the questionnaire as Supplementary table 1. 

4. The major utility of this work is its potential to serve as a general view of the logistic set-up used for this study. A significant concern that should be addressed in the discussion is: How likely are these results, collected as part of an infectious disease surveillance study in the middle of a pandemic, likely to generalize to studies post-pandemic? In other words, are the participation rates for the demographic categories mentioned, skewed by the overwhelming desire to stay safe during the pandemic? It is quite possible that participation rates or viewpoints may change outside the current environment. This should at least be addressed in the discussion.

We thank the reviewer for bringing up this important point. We clarified the complexities, constraints, and benefits of conducting such population studies during a pandemic, and how they might change in a non-pandemic setting:

“Participants were certainly influenced by the climate surrounding the pandemic and the public health measures implemented, such as stay-at-home orders and social distancing, and thus were likely to have been more inclined to partake in a remote surveillance study. While the incentive to do so in a non-pandemic world could be reduced, the low amount of sample needed and reduced pain in comparison to traditional venipuncture approaches are benefits that might retain interest in participating in future remote studies. Importantly, such remote design would facilitate collection of samples in very large scale (eg nationwide) without the need for very large teams for community engagement and sample collection, therefore being potentially useful in effectively integrating public health pathogen surveillance (ie “peacetime” surveillance) into day-to-day practices of community and individual health, hopefully viewed as a worthy cause in which individuals participate to avert future pandemics.”

---

## [Editor Report · Decision Letter 1]

29 Sep 2021

Development of at-home sample collection logistics for large-scale seroprevalence studies

PONE-D-21-24248R1

Dear Dr. Aatresh,

We’re pleased to inform you that your manuscript has been judged scientifically suitable for publication and will be formally accepted for publication once it meets all outstanding technical requirements.

Kind regards,

James Drummond

Academic Editor

PLOS ONE

Additional Editor Comments (optional):

Thank you and your authors for addressing the reviewers concerns.
---

## [Editor Report · Acceptance letter]

21 Oct 2021

PONE-D-21-24248R1 

Development of at-home sample collection logistics for large-scale seroprevalence studies 

Dear Dr. Aatresh:

I'm pleased to inform you that your manuscript has been deemed suitable for publication in PLOS ONE. Congratulations! Your manuscript is now with our production department. 

Kind regards, 

on behalf of

Dr. James Drummond 

Academic Editor

PLOS ONE